# The Correlations between Horizontal and Vertical Peripheral Refractions and Human Eye Shape Using Magnetic Resonance Imaging in Highly Myopic Eyes

**DOI:** 10.3390/healthcare9080966

**Published:** 2021-07-30

**Authors:** Hui-Ying Kuo, John Ching-Jen Hsiao, Jing-Jie Chen, Chi-Hung Lee, Chun-Chao Chuang, Han-Yin Sun

**Affiliations:** 1Department of Optometry, Chung Shan Medical University, Taichung 402, Taiwan; evekuo@csmu.edu.tw (H.-Y.K.); hsiao@csmu.edu.tw (J.C.-J.H.); frankchen@iki.com.tw (J.-J.C.); 2Department of Ophthalmology, Chung Shan Medical University Hospital, Taichung 402, Taiwan; 3Department of Electrical Engineering, Feng Chia University, Taichung 407, Taiwan; chihlee@fcu.edu.tw; 4Department of Medical Imaging and Radiological Sciences, Chung Shan Medical University, Taichung 402, Taiwan; jimchao@csmu.edu.tw

**Keywords:** high myopia, peripheral refraction, magnetic resonance imaging, ocular shape

## Abstract

The aim of this study was to determine the relationship between relative peripheral refraction and retinal shape by 2-D magnetic resonance imaging in high myopes. Thirty-five young adults aged 20 to 30 years participated in this study with 16 high myopes (spherical equivalent < −6.00 D) and 19 emmetropes (+0.50 to −0.50 D). An open field autorefractor was used to measure refractions from the center out to 60° in the horizontal meridian and out to around 20° in the vertical meridian, with a step of 3 degrees. Axial length was measured by using A-scan ultrasonography. In addition, images of axial, sagittal, and tangential sections were obtained using 2-D magnetic resonance imaging. The highly myopic group had a significantly relative peripheral hyperopic refraction and showed a prolate ocular shape compared to the emmetropic group. The highly myopic group had relative peripheral hyperopic refraction and showed a prolate ocular form. Significant differences in the ratios of height/axial (1.01 ± 0.02 vs. 0.94 ± 0.03) and width/axial (0.99 ± 0.17 vs. 0.93 ± 0.04) were found from the MRI images between the emmetropic and the highly myopic eyes (*p* < 0.001). There was a negative correlation between the retina’s curvature and relative peripheral refraction for both temporal (Pearson r = −0.459; *p* < 0.01) and nasal (Pearson r = −0.277; *p* = 0.011) retina. For the highly myopic eyes, the amount of peripheral hyperopic defocus is correlated to its ocular shape deformation. This could be the first study investigating the relationship between peripheral refraction and ocular dimension in high myopes, and it is hoped to provide useful knowledge of how the development of myopia changes human eye shape.

## 1. Introduction

Since myopia has been reached an epidemic level in East-South Asia, the prevalence of high myopia (>5 D of myopia) was apparent in the first studies from Taiwan [1,2]. In the past few decades, the prevalence of high myopia has increased by as much as 10-fold, suggesting that the percentage of myopes who become highly myopic has increased [3]. In a recent meta-analysis of 145 studies, there are 1950 million myopes (28.3% of the global population) and 277 million high myopes (4.0% of the global population) around the world, respectively, and these numbers will increase rapidly and become twice or more by 2050 [4]. Since eye shape has been focusing as a possible biomarker and links to the development of myopia, for example, longer axial elongation accompanying myopia progression has been proved while the emmetropes remained [5,6,7]. The shape of the eye tends to become steeper and flatten away from the posterior pole with increased myopia [8]. The stretching that occurs in the peripheral retina parallel to the visual axis manifests in highly myopic eyes and can cause various ocular complications [5]. In young children, it was found that the anterior ocular surface, rather than the eye volume, was significantly associated with the spherical equivalent [9]. A prolate shape in high myopic eyes conformed to the axial expansion model, whereas non-myopic eyes followed the global expansion model.

These results suggested that during emmetropization, the shape of the globe changes globally and the eyes tend to become prolate with increased spherical equivalent after emmetropization completes. It is still unclear that what exactly the mechanism is to make eyes elongated with developing myopia, even though there are a large number of studies demonstrating that the basis of the relationship between eye shape and peripheral refraction [10,11,12]. However, it has been suggested that relative peripheral hyperopia could be a consequence rather than the cause of myopia [13,14,15]. During the development of myopia, the more relative peripheral hyperopia exhibits, the steeper and more prolate the retina is suggested to be. Several experiments have shown that optically-imposed changes in the eye’s refractive state predictably altered refractive development in a manner that reduces the optically-imposed error [16,17,18]. Thus, the eye can alter the rate of vitreous chamber elongation to restore emmetropia. Visual signals that accelerate axial growth were mediated by mechanisms that operate in a local and regionally selective manner [19]. Peripheral refraction stimulates the elongation of the eye may be due to the fact that the defocus signal is stronger in the peripheral retina than in its center. However, how the signals produced by myopic defocus were integrated across the retina is relatively little known. In animal models, it has been found that positive lenses which induced relative myopic defocus over half the retina produced hyperopic shifts but were restricted to the treated half of the eye. The peripheral retina could function alone, and an intact fovea was not essential for emmetropization [20]. It is hypothesized that the eye has an increased risk of shifting to myopia if the relative peripheral defocus is hyperopic as the focal plane is placed behind the peripheral retina [21]. More evidence is required to discover the role of peripheral refraction in vitreous chamber elongation and following ocular expansion models of the globe.

The retina’s morphology has been reported to be associated with refractive development and is considered important in myopiogenesis [6]. It has been found in adult eyes that ocular shape is associated with refractive error; myopic eyes tend to be prolate, whereas hyperopic eyes tend to be oblate. Mutti et al. showed that myopic children had more significant relative hyperopia in the periphery (a prolate shape of the eye) than relative peripheral myopia as in emmetropic and hyperopic eyes (an oblate form of the eye) [10]. The results suggested that myopic children’s eyes were both elongated and distorted into a prolate shape; however, the association between a prolate ocular form and the peripheral refraction was not addressed. Magnetic resonance imaging (MRI) has provided a more direct manner to measure the shapes of highly myopic eyes and makes it possible for us to investigate the entire globe shape and provide comprehensive biometric information for its morphometric analysis [8,22,23]. Moriyama et al. suggested that the globe’s quantitative assessments were useful in determining the pattern of eye shape deformity caused by pathologic myopia, of which the emmetropic eyes were symmetrical horizontally and sagittally without deformity, whereas the posterior pole’s shape was pointed in 45.7% and blunted in 54.3% in highly myopic eyes showing that a blunt posterior surface accompanied by progressive high myopia [24]. A less oblate posterior eye shape was also found to be associated with smaller spherical equivalent and longer axial length using MRI examinations. However, the correlation of corresponding peripheral refraction to its retinal curvature has not been studied so far.

The correlations between the horizontally and vertically relative peripheral refractions and the retinal curvature of the corresponding regions measured using magnetic resonance imaging were investigated in this study. The ocular dimensions (including the axial, tangential, and sagittal lengths) were measured using magnetic resonance imaging, and the retinal curvature of the emmetropic and highly myopic eyes was determined by using the MATLAB software.

## 2. Materials and Methods

### 2.1. Participants

This study adhered to the Declaration of Helsinki’s tenets and received ethical clearance from the Human Research Ethics Committee of Chung Shan Medical University Hospital (CSMUH No: CS13260). Informed consent was obtained from all the subjects after a clear explanation of the experimental procedures. Thirty-five Asian young adults aged 20 to 30 years (mean: 23 ± 3 years old) were recruited, including 22 females and 13 males. Subjective refraction was applied to check their refractive errors before the peripheral refraction measurement and magnetic resonance images. The participants included 16 emmetropic (spherical equivalent: +0.50 to −0.50 D) and 19 high myopic (<−6.00 D) age-matched subjects. The inclusion criteria were corrected/uncorrected visual acuity of 20/20 or better and less than 1.00 D of astigmatism. The subjects were excluded if they had any ocular pathology such as strabismus and amblyopia or any ocular surgery and any orthokeratology lens history.

### 2.2. Peripheral Refraction Measurement

Spherical equivalents and peripheral refractions were measured using an open-field autorefractor (Shin-Nippon SRW5000; Shin-Nippon Commerce Inc., Tokyo, Japan). Non-cycloplegic refractions were measured along the horizontal meridian in 3° steps out to 60° and out to 9° of the superior and 12° of the inferior visual fields. Central refraction (CR) was determined while the subjects were fixating at a Snellen 0.05 E letter target, which was placed straight ahead at a viewing distance of 4 m. During PR measurements, the subjects were seated with the head stabilized in a chin rest so that the testing eye was aligned with the high contrast letter E target placed at 6 m in the peripheral field. Five measurements were taken at each eccentricity, and the average was used for data analysis.

Conventional sphero-cylindrical refractive error (S/C × θ, where the cylinder was in negative form) measured by the open-field autorefractor was transposed into power vector form (M, J_0_, J_45_) in which M equals the mean spherical equivalent power, J_0_ represents 90 to 180-degree astigmatic component), and J_45_ represents 45 to 135-degree astigmatic component for data analysis, where:M = S + C/2J_0_ = −2(Cylinder/2) × cos(2θ)J_45_ = −2(Cylinder/2) × sin(2θ)(1)

Relative peripheral astigmatic refraction was consisted of tangential and sagittal power errors, and the sagittal refractive error (FS′) and tangential refractive error (FT′) were also calculated for data analysis. Relative peripheral refraction (RPR) was calculated as the difference between the absolute refraction measured on-axis and that measured at each peripheral eccentricity. Consequently, a hyperopic RPR is represented as positive values in the results, while negative values represent a myopic RPR.
F′_T_ = M + J_0_F′_S_ = M − J_0_(2)

### 2.3. MRI Images Collection

The magnetic resonance images were obtained using a 1.5 T clinical system with a 7.5 cm receive-only surface coil while the contralateral eye was being occluded. A T1-weighted fast spin-echo sequence (an inversion time of 1280 ms and a short interval of 3000 ms; the repetition time, the echo time, and the flip angle were 2.5 ms, 4.55 ms and 16°, respectively) was used to acquire the MR images of tangential axial (horizontal through the middle of the eye) and sagittal axial (vertical through the visual axis) sections. Sagittal fast spin-echo images were taken with fat suppression to minimize the chemical shift artifact of the sclera’s inferior region. The scan time was 2 min and 50 s, and resulted in a field of view of 40 × 46 × 38 mm. Each magnetic resonance image had an in-plane resolution of 0.264 mm/pixel with 320 × 320 pixels. The total MRI examination time taken was around 10~12 min for each subject. A fixation target (a 20/40 E letter) was placed straight ahead of the eye’s visual axis with a viewing distance of 35 cm. Complete data of MRI images were obtained only for 50 out of total 70 eyes (emmetropic: 24 eyes; highly myopic: 26 eyes) due to slight eye movement or cyclotorsion would lead to unclear MRI images, and thus the results were not able to be calculated and converted into the curvature of the retina. Even though a near fixation target was applied to maintain the subjects’ primary gaze, a blurred vision may occur in some of the subjects since lens wearing was not allowed during the MRI examination, and in this case, motion artifacts cannot be fully excluded.

The data of MRI were analyzed with custom-written software MATLAB (MATLAB 8.0 program). The 2-D curvature of the retina for both the tangential and sagittal sections was determined for the corresponding horizontal and vertical meridians measured in peripheral refraction. The MRI pictures of one of the subjects are shown in Figure 1a,b to help explain how the retinal curvature was determined. The axial length measured with MRIs was compared with A-scan ultrasound measurement (Axis-II A-scan, Quantel Medical, Rockwall, TX, USA) during a distant fixation in the same eye for each participant to confirm the dimension of the eyeball as both the measurements were similar. Various biometric parameters were measured and determined automatically based on the collected MRI images by MATLAB [25] and Rhinoceros [26].

The sequence of data analysis was as follows: (1) an MR image was converted into grayscale using MATLAB software (see Figure 1a,b); (2) according to the distribution of pixel levels, the threshold of the image matrix was tuned to determine the boundaries of cornea, lens, and retina (see Figure 1c; for example, the grayscale values which represent the regions of the retina and the vitreous were above 0.35 and below 0.2, respectively, in the same MRI pictures. Therefore, the threshold value was set to be 0.275 for clearly identifying the shape of retina, and (3) the coordinates of the boundaries and their corresponding dimensions were illustrated using Rhinoceros (see Figure 1d). In this case, since the retinal profile consisted of a thickness of a few pixels, the possible radius value R was distributed within a range, rather than a single value, such as the values 10.64, 11.01, and 11.88 mm shown in Figure 1d, which were applied for data analysis of retinal shape simulation (see Section 2.4).

### 2.4. Data Analysis

Two different types of software, MATLAB and Rhinoceros, were used to find an association between independent and dependent variables that, when graphed, produces a straight line, plane, or curve. As adopted by the previous study [27], the regression procedures found the equation that most closely describes, or fits, the actual data of MRI images and the results from A-scan ultrasonography, which can be used to predict/simulate the curvature of ocular shape in the emmetropic and highly myopic eyes.

Statistical analyses were performed with commercially available software (SPSS ver. 11.0; SPSS Inc., Chicago, IL, USA). The total average and mean peripheral refraction measurements (M, J_0_, and J_45_) between high myopic and emmetropic groups were compared by using an independent *t*-test. Correlations between peripheral refraction and the curvature of ocular shape at the corresponding retinal regions were examined by linear regression analysis and expressed as the Pearson coefficient of correlation (r). *p* < 0.05 was considered statistically significant.

## 3. Results

Thirty-five participants were divided into two groups: 16 subjects (32 eyes) were emmetropes and 19 subjects (38 eyes) were high myopes. The descriptive data were shown in Table 1. No significant difference in age was found as the both groups had similar mean ages, however, statistically significant differences were found in AL (t_66_ = −13.52, *p* < 0.001), SE (t_42.85_ = 34.46, *p* < 0.001), FT′ (t_44.53_ = 32.03, *p* < 0.001) and FS′ (t_42.46_ = 33.26, *p* < 0.001).

In comparing horizontal peripheral refractions between the two groups, statistically significant differences were found in spherical equivalents measured horizontally at all the eccentric angles (a step of 3°) for both the nasal and the temporal visual fields (*p* < 0.001). The high myopic group showed relatively hyperopic refraction for all the horizontally measured angles (Figure 2a). Significant differences were also found in FT′ power (*p* < 0.01) measured horizontally at all of the eccentricities (Figure 2b); but in FS′ power, the differences were not significant for T03 (*p* = 0.067) and N03 (*p* = 0.174) while the remain eccentricities were significant (*p* < 0.01) (Figure 2c).

For the vertical peripheral refraction, the high myopic group showed relatively hyperopic refraction compared to the emmetropic group (*p* < 0.001), and the differences became manifest as the eccentricity increased (Figure 3a). Significant differences were also found in vertical FT′ power (*p* < 0.01) and FS′ power (*p* < 0.01) for all of the eccentric angles (Figure 3b,c).

The high myopic eyes were larger in axial length, horizontal width, and vertical height by analyzing the collected magnetic resonance images. Statistically significant differences were found between the high myopic group and the emmetropic group in axial length (26.15 ± 1.05 mm vs. 23.40 ± 0.68 mm; t_26_ = −7.90, *p* < 0.001), in horizontal width (24.41 ± 1.22 mm vs. 23.19 ± 0.90 mm; t_26_ = −2.93, *p* < 0.001), and in vertical height (24.54 ± 0.91 mm vs. 23.53 ± 0.82 mm; t_26_ = −2.99, *p* < 0.001). The ratios of the emmetropic eyes and the highly myopic eyes were also calculated. There were significant differences in the ratios of width/axial and height/axial between the highly myopic group and the control group (*p* < 0.001), but not for the ratio of height/width. For the emmetropic group, the mean ratios of width/axial, height/axial, and height/width were 0.99 ± 0.17, 1.01 ± 0.02, and 1.02 ± 0.02, respectively. The mean ratios of width/axial, height/axial, and height/width of the high myopic group were 0.93 ± 0.04, 0.94 ± 0.03, and 1.01 ± 0.04 (Table 2).

In the emmetropic group, the peripheral refractions were not significantly correlated with the retina’s curvature at any of the four visual fields. However, statistically significant negative correlations between the curvature of the retina and peripheral refraction in the high myopic group were found in the temporal area of the retina (Pearson coefficient r = −0.459; *p* < 0.01) (see Figure 4a); and in the nasal retina (Pearson coefficient r = −0.277; *p* = 0.011) (see Figure 4b). There were no significant correlations between the retina’s curvature and the corresponding refractive errors in the superior and inferior retina (Figure 4c,d). For the high myopic group, the peripheral refraction was more hyperopic, and the curvature of the retina became flatter in the horizontal meridian.

## 4. Discussion

A more hyperopic relative peripheral refraction in both the horizontal and vertical retina was observed in the high myopic eyes compared to the emmetropic eyes, which is consistent with the previous studies [10,28]. Peripheral hyperopic refraction is considered to be responsible for myopia development in a number of animal models. The eye’s visually guided growth mechanism in the retina attempts to compensate with axial elongation for the imposed peripheral defocus even though the central image is correctly focused [29,30]. As pointed out in previous studies, high myopes showed higher amounts of relative hyperopic refractive error mainly along the horizontal meridian compared to emmetropes. By performing open field autorefraction measurements along both horizontal and vertical meridians, Atchison et al. found that myopic eyes showed relative hyperopia along the horizontal meridian but relative myopia along the vertical meridian [31]. Similarly, Berntse et al. reported that myopic children were relatively hyperopic in the horizontal periphery and relative myopic in the vertical periphery [32]. However, a cross-sectional study conducted by Ehsaei et al. showed that myopic eyes had a relative hyperopic shift in the periphery along all four measured meridians [33]. In the present study, relative peripheral hyperopia was also found in the highly myopic eyes (>6 D of myopia). In contrast, the emmetropic eyes showed relative peripheral myopia along both the horizontal and vertical meridians. The multi-variables that may cause the different observations in peripheral refraction as myopia increased, such as ethnicity and degree of myopia, should be included for data analysis.

Although high myopic eyes expanded in all directions relative to emmetropic eyes, they were elongated more in the axial direction (the emmetropic group: 23.40 ± 0.68 mm; the high myopic group: 26.15 ± 1.05 mm) than in the vertical dimension and even less in the horizontal dimension. The ratio of horizontal width to the central axial length of the emmetropic group is 0.99, the ratio of vertical height to central axial length is 1.01, and the ratio of vertical height to horizontal width is 1.02, which means the three-dimensional shapes of the emmetropic eyeballs was a vertically elongated ellipse with longer vertical length and shorter horizontal length. In contrast, the ratio of horizontal width to the central axial length of high myopia is 0.93, the ratio of vertical height to central axial length is 0.94, the ratio of horizontal width to the vertical height of high myopia is 1.01, showing that the high myopic eyes were longitudinally elongated ellipse with a slightly larger dimension in height than in width. These findings were consistent with the results of Atchinson et al.’s study that myopic eyes became much more extensive in all dimensions as the magnitude of myopia increased, but more so in length (0.35 mm/D) than in height (0.19 mm/D) or width (0.10 mm/D) in the participants aged 18 to 36 years [8]. Three mechanisms of stretching in myopia have been discussed: equatorial elongation, global elongation, and posterior polar elongation. The authors believed that their model is intermediate between the equatorial and global expansion models with a differential growth rate in different meridians, even though the individual variations could exist in any single eye.

Ishii et al. recruited 105 young Asian children aged one month to 19 years and conducted eye shape analysis based on T2-weighted MRI images. The results showed that there was a significant correlation between an oblate-to-prolate change and spherical equivalent among the children aged seven years or older, which suggested that the eyes tend to become prolate as their myopia developed after the emmetropization is done [34]. In another study on the shape of posterior vitreous chamber imaged by T2-weighted 3D MRI in fifty-five young adults, however, it was reported that the prolate ellipse posterior chamber shapes were rarely found in myopic eyes [35]. The previous studies indicate that the ocular dimensions increase in all directions as myopia increases in adult eyes. The increase in axial elongation is significantly more obvious than the increase in width and height, and therefore resulting in a prolate eye shape as myopia develops [9,35,36]. It was also evident that different races could affect the ocular shape of the human eye, with a less oblate shape in Chinese subjects [27]. In the present study, a prolate shape of the eye was found in the highly myopic Asian adults with a larger expansion more in width than in height.

The horizontal retina deformation has a smaller central retinal curvature, and then the retina expands outwards in the periphery. The absolute peripheral refractions and curvatures of the high myopic group were found to be negatively correlated horizontally (temporal side: Pearson r = −0.459; *p* < 0.01; nasal side: Pearson r = −0.277; *p* = 0.011), which showed that the high myopic subject’s refraction tends to be more hyperopic as the deviation angle increases, and the corresponding retinal curvature decreases. The results were only found in the horizontal meridian, but not in the vertical meridian, consistent with the findings of Stone et al. [37] and Smith III et al. [38]. The experiment using the chick model showed that a deprivation of the temporal retina generated more central myopia than depriving the nasal retina [37], which suggested that eye growth is locally controlled within the eye and can be influenced by the environment [39]. Prolonged near work and very limited outdoor exposure has been considered to contribute to current pandemic of myopia in Asian children. A recent animal experiment has demonstrated that natural light exposure provides a protective effect against myopia development by more hyperopic refraction and shorter vitreous chamber depth [40]. Since the mechanism of emmetropization is contained within the eye and peripheral defocus can alter refractive development [16], the defocus experienced by the peripheral retina would be expected to drive eye growth [13]. And therefore, the corresponding deformation that occurred in the retina of the high myopes, in particular the temporal region, may account for the significant hyperopic peripheral refraction found in the present study. The retinal shape measured by MRI as a characteristic of refractive error in high myopes and its correlation to peripheral refraction were evaluated. It is possible that peripheral refraction measurement can be used in predicting the deformation of the globe, particularly for high myopes because of its consistency with other clinical imaging techniques.

## 5. Conclusions

The peripheral hyperopic defocus is correlated with the retina’s shape deformation as a higher degree of myopia develops. The changes in the curvature of the retina in the highly myopic eyes were manifest in the horizontal meridian than the vertical meridian and particular for the temporal retina. This study provided clear evidence between peripheral refraction and ocular shape from MRI measurement in the highly myopic human eyes.

## Figures and Tables

**Figure 1 healthcare-09-00966-f001:**
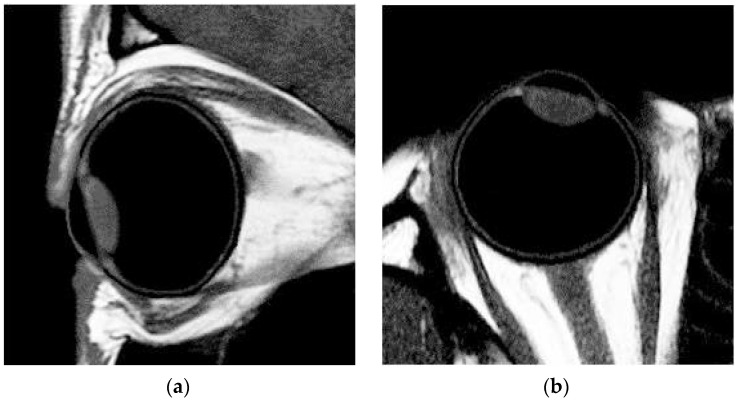
(**a**) Tangential axial MR image of one of the subjects. (**b**) Sagittal axial MR image of one of the subjects. (**c**) Various biometric parameters were measured using magnetic resonance imaging. (**d**) The curvature of the retina was calculated using the Rhinoceros.

**Figure 2 healthcare-09-00966-f002:**
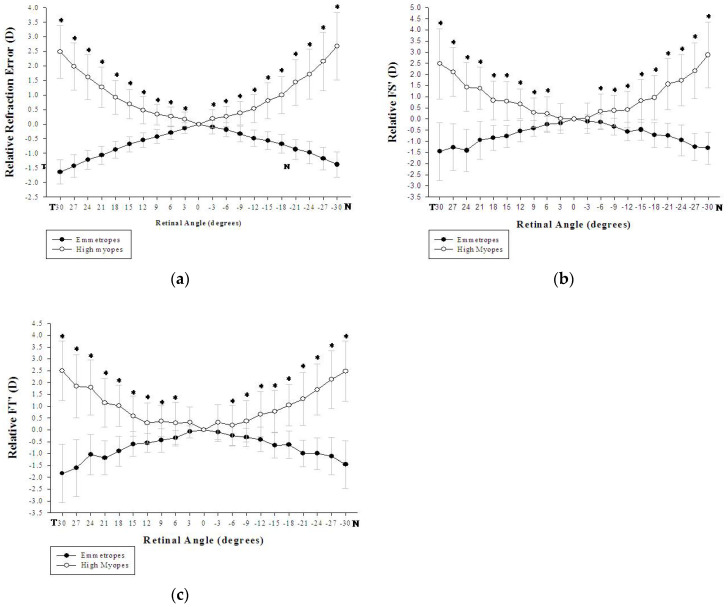
There were significant differences in the horizontally measured relative peripheral refraction between the emmetropic and high myopic groups in (**a**) Spherical equivalents at all of the eccentricities (* *p* < 0.001); (**b**) Tangential refractive error (FT′) at all of the eccentricities (* *p* < 0.01); and (**c**) Sagittal refractive error (FS′) (* *p* < 0.01) except for T03 (* *p* = 0.067) and N03 (* *p* = 0.174).

**Figure 3 healthcare-09-00966-f003:**
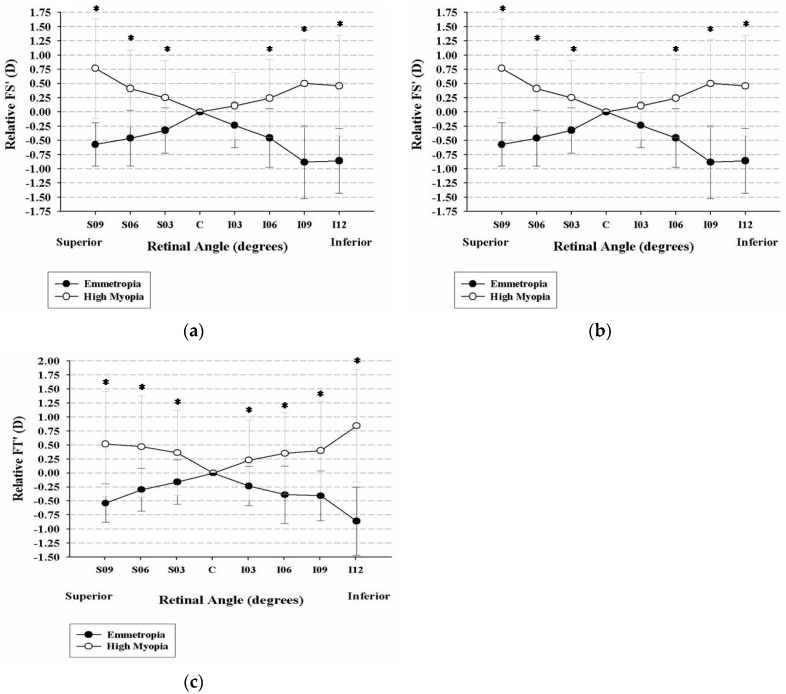
There were significant differences in the vertically measured peripheral refraction between the emmetropic and high myopic groups in (**a**) Spherical equivalents at all of the eccentricities (* *p* < 0.001); (**b**) Tangential refractive error (FT′) (* *p* < 0.01) at all of the eccentricities; (**c**) Sagittal refractive error (FS′) at all the eccentric angles (* *p* < 0.01).

**Figure 4 healthcare-09-00966-f004:**
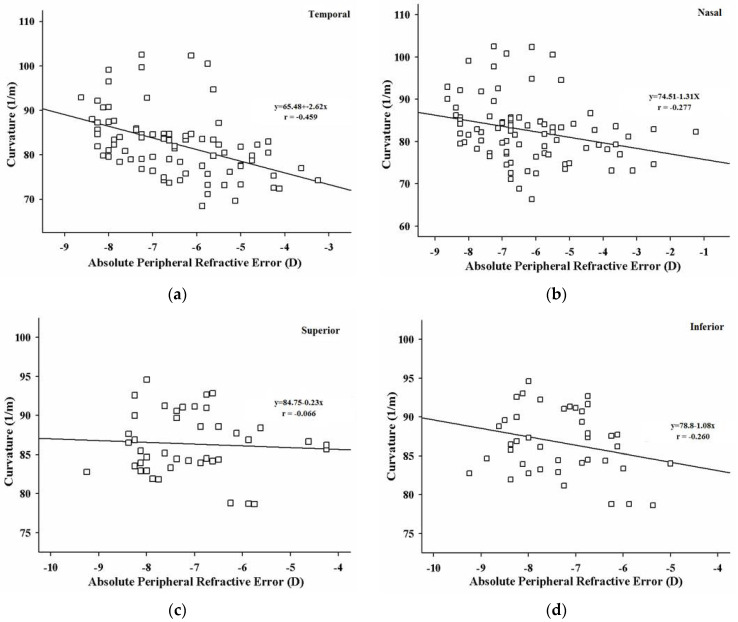
The graphs showed the negative correlations between the retina’s curvature and the peripheral refraction in the highly myopic group. (**a**) There was a significant correlation between the retina’s curvature and the peripheral refraction in the retina’s temporal area (r = −0.459; *p* < 0.01). (**b**) There was a significant correlation between the retina’s curvature and the peripheral refraction in the nasal area (r = −0.277; *p* = 0.011). (**c**) No significant correlation was found in the superior retina. (**d**) No significant correlation was found in the inferior retina.

**Table 1 healthcare-09-00966-t001:** Descriptive data of the subjects enrolled in the study.

	Age	N (Eyes)	AL (mm)	SE (D)	FT′ (D)	FS′ (D)
Emmetropes	23 ± 3	32	23.70 ± 0.65	−0.03 ± 0.36	−0.04 ± 0.44	−0.02 ± 0.36
High Myopes	23 ± 3	38	26.41 ± 1.19	−8.08 ± 1.40	−8.07 ± 1.48	−8.09 ± 1.46

AL: axial length, SE: spherical equivalence, FT′: tangential refractive error, FS′: sagittal refractive error.

**Table 2 healthcare-09-00966-t002:** The mean axial length, height, width and the dimensional ratios of the emmetropic and highly myopic groups.

	Emmetropes (n = 12)	High Myopes (n = 16)	*p* Value
Central Axial (mm)	23.40 ± 0.68	26.15 ± 1.05	<0.001
Vertical Height (mm)	23.58 ± 0.61	24.59 ± 0.87	0.006
Horizontal Width (mm)	23.19 ± 0.90	24.41 ± 1.21	0.007
Height: Axial	1.01 ± 0.02	0.94 ± 0.03	<0.001
Width: Axial	0.99 ± 0.17	0.93 ± 0.04	<0.001
Height: Width	1.02 ± 0.02	1.01 ± 0.04	0.485

## Data Availability

The raw data supporting the conclusions of this article will be made available by the authors.

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
