# Peer review of "The Correlations between Horizontal and Vertical Peripheral Refractions and Human Eye Shape Using Magnetic Resonance Imaging in Highly Myopic Eyes"

_healthcare, 2021, doi:10.3390/healthcare9080966_

Round 1

Reviewer 1 Report

The article The correlations between horizontal and vertical peripheral refractions and human eye shape using magnetic resonance imaging in highly myopic eyes is well structured, although there are some aspects that need to be improved and some corrections and/or observations that the authors should consider in order to improve the article:

Authors should clarify these aspects.

L 29-31: This could be the first study investigating the relationship between peripheral refraction and ocular dimension in high myopes, and it is hoped to provide useful knowledge of how the development of myopia changes human eye shape.

Does myopia change the eye shape or is it the change in the shape of the eye that gives rise to myopia?

-L 254-257: Peripheral hyperopic refraction is responsible for myopia development in a number of animal models. The eye's visually guided growth mechanism in the retina attempts to compensate with axial elongation for the imposed peripheral defocus even though the central image is correctly focused.

What physiological process has as a consequence the change in the shape of the eye in the myopes in front of the eye of the emmetropes gives rise to the Peripheral hyperopic refraction?

I think it is not correct to say that Peripheral hyperopic refraction is responsible for myopia development but is a consequence of the process of myopic eye. (L57-58: it has been suggested that relative peripheral hyperopia could be a consequence rather than the cause of myopia [13- 15])

The authors should clarify this point in the discussion section.

-There are multiple articles that relate the lack of natural light in the first years of life to the current pandemic of myopia. I think the authors should refer to this possibility. The authors only make a brief mention (L317-318). I believe that more mention should be made of this possibility.

-L 304. It was also evident that different races could affect the ocular shape of the human eye, with a less oblate shape in Chinese subjects.

It would be interesting to specify whether the 35 participants are of the same race and anatomical characteristics.

Reviewer 2 Report

Review MRI in highly myopic eyes

  1. Overall – very interesting subject relevant especially in Asian populations. A novel approach.
  2. English proofreading and correction of some phrasal mistakes is necessary. There are places where the text is difficult to follow due to poor translation I guess.
  3. 1 . Please correct a, b, subtitles in the legend.
  4. Authors should give some details in brief on Matlab and Rhinoceros software.
  5. Provide explanation for FT and FS.
  6. Table 1 . Provide the legend for abbreviations.
  7. Discussion: please state clearly how your work is different from other publications in that field.
